# An oocyte meiotic midbody cap is required for developmental competence in mice

Gyu Ik Jung [1,2], Daniela Londoño-Vásquez[3], Sungjin Park [4], Ahna R. Skop [4], Ahmed Z. Balboula [3] & Karen Schindler [1,2] ✉

Embryo development depends upon maternally derived materials. Mammalian oocytes undergo extreme asymmetric cytokinesis events, producing one large egg and two small polar bodies. During cytokinesis in somatic cells, the midbody and subsequent assembly of the midbody remnant, a signaling organelle containing RNAs, transcription factors and translation machinery, is thought to influence cellular function or fate. The role of the midbody and midbody remnant in gametes, in particular, oocytes, remains unclear. Here, we examined the formation and function of meiotic midbodies (mMB) and mMB remnants using mouse oocytes and demonstrate that mMBs have a specialized cap structure that is orientated toward polar bodies. We show that that mMBs are translationally active, and that mMB caps are required to retain nascent proteins in eggs. We propose that this specialized mMB cap maintains genetic factors in eggs allowing for full developmental competency.

Oocytes, the gametes derived from ovaries, undergo a maturation process that couples the completion of meiosis I with acquisition of developmental competence essential to support preimplantation embryogenesis. During meiotic maturation, eggs acquire developmental competence by rearranging organelles, degrading and translating maternal mRNAs, and erasing epigenetic modifications[1]. Importantly, after fertilization, early embryo development depends on proteins synthesized in the egg.

During meiosis I completion, oocytes segregate homologous chromosomes and undergo an asymmetric cytokinesis. This asymmetric cytokinesis event results in a large egg cell and a non-functional cell called a polar body (PB) (Fig. 1a). The completion of meiosis I and extrusion of PBs signifies maturation of oocytes into eggs[2]. In somatic cells, a transient organelle called a midbody (MB) forms between dividing cells during early and late Telophase. Cytokinesis in somatic cells not only involves separation into two equally sized daughter cells, but, upon abscission, a large extracellular vesicle called the midbody remnant (MBR) forms[3–5] (Fig. 1a). When abscission occurs on both sides of the MB, MBs are released extracellularly and recipient cells can phagocytose them[6–8]. When phagocytosed by cancer and stem cells, these MBRs appear to regulate tumorigenicity and stemness,

respectively, suggesting that MBR uptake has cell type-specific and fate determining effects[9–11]. The formation of MBs and MBRs in oocytes is unclear (Fig. 1a).

How MBs and MBRs influence cellular behavior in both somatic and germ cells is not well understood. Recent work attributes this ability to the RNAs that are potentially translated or used as templates to inhibit gene function after MBRs are internalized[12]. In somatic cells, MBs and MBRs are enriched with translational machinery and specific transcripts that are recruited via their 3' untranslated region sequence, providing further explanation of how MBRs could act as signaling organelles[12,13]. Because oocytes must produce proteins critical for successful meiosis and early embryogenesis, we hypothesized that MBs would locally translate proteins. Furthermore, we hypothesized that the egg would have a mechanism to retain these proteins by preventing their escape into the PB, a mechanism which could be critical to produce a developmentally competent egg.

Here, we report the presence of a meiotic MB (mMB) in mouse oocytes and describe a mMB cap-like sub-structure that contains Centralspindlin complex proteins MKLP1 and RACGAP. We report that the mMB is translationally competent and provide evidence that the mMB cap is the boundary for translation between the egg and PB.

[1]Department of Genetics, Rutgers, The State University of New Jersey, Piscataway, NJ, USA. [2]Human Genetics Institute of New Jersey, Piscataway, NJ, USA. [3]Animal Sciences Research Center, University of Missouri, Columbia, MO, USA. [4]Laboratory of Genetics, University of Wisconsin-Madison, Madison, WI, USA. ✉e-mail: ks804@hginj.rutgers.edu

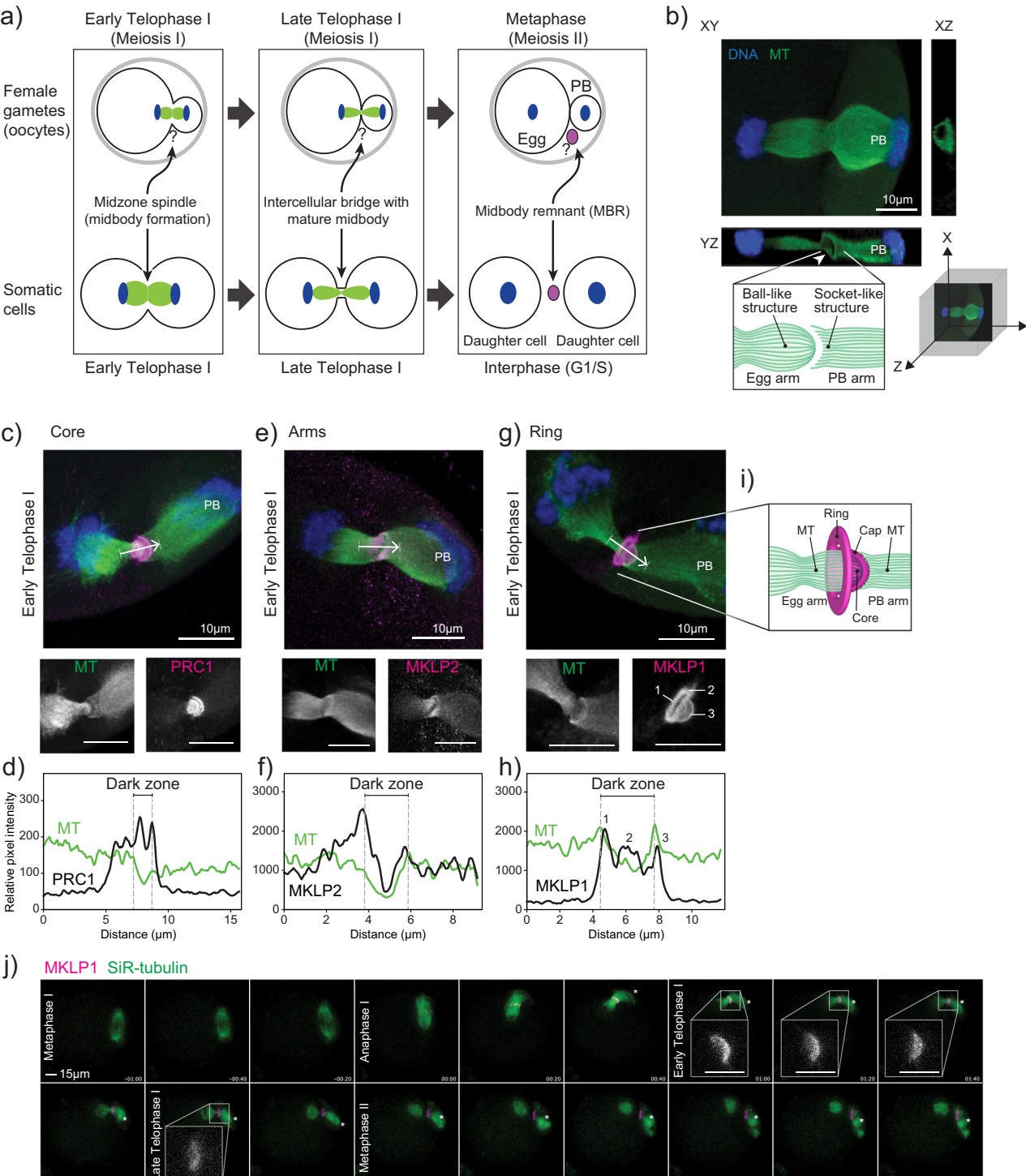

**Fig. 1 | The meiotic midbody of mouse oocytes contains a cap sub-structure.**
**a** Schematic depicting the distinct processes of meiosis I cytokinesis in oocytes and mitotic cytokinesis. Question marks refer to unknown biology in oocytes.
**b** Representative image of a mouse oocyte early Telophase I with views at XY, XZ and YZ planes. White arrowhead in YZ plane highlights the tubulin asymmetry (ball-like structure and socket-like structure); polar body (PB). Below confocal images are a 3D coordinate system and schematic of the observations.
**c, e, g** Representative z-plane projected confocal images showing localization of representative markers for the three main midbody regions (magenta in merge; PRC1, MKLP2 and MKLP1) relative to microtubules (green in merge; alpha-tubulin) and chromosomes (blue in merge; DAPI). White arrows indicate direction of line

scan plots in **d**, **f**, and **h**. The numbers in the MKLP1 zoom panel correspond to the three peaks in the line scan in (**h**). **d, f, h** Intensity line scan plots for microtubules (green) and corresponding protein (black) of images in **c**, **e**, **g**. Gray dotted lines demark the beginning and end of the midbody dark zones. **i**) Schematic representation of the meiotic midbody and cap in oocytes. **j**) Representative still images from confocal live-cell imaging of oocytes, expressing MKLP1-GFP (magenta) and incubated with SiR-tubulin (green). Stages of meiosis are labeled. Timepoint 00:00 represents the beginning of Anaphase I onset. Four zooms of early and late Telophase I time points highlight the cap formation and regression. All experiments were conducted three times. Asterisk marks the PB. Zoom scale bars = 5 μm.

Importantly, we demonstrate that this cap contributes to the full developmental competence of eggs by preventing maternal proteins from leaking into the first PB prior to abscission. Taken together, our findings highlight a mechanism by which a meiotic cell modifies mitotic machinery to provide developmental benefits for egg and embryo quality.

## Results and discussion

### Mouse oocyte meiotic midbodies contain a cap sub-structure

Because the genesis and morphology of mouse oocyte midbodies were unknown (Fig. 1a), we first assessed the morphology of mMBs. Confocal imaging of anti-tubulin-stained early Telophase I-staged oocytes revealed that the microtubules at the spindle midzone had a unique structure: the microtubules on the maturing oocyte (or egg) side always terminated in a ball-like structure (left) and the spindle microtubules on the PB side always terminated in a socket-like structure (right) (Fig. 1b). To our knowledge, this meiotic Telophase I microtubule asymmetry does not exist in mitotic Telophase. For ease of orientation, the egg side of all subsequent images will be presented on the left and the PB side will be presented on the right side and labeled.

To further investigate the morphology of mMBs, we identified the three landmark regions described in mitotic MBs (core, arms, ring)[3–5] by immunofluorescence to detect protein regulator of cytokinesis 1 (PRC1) and mitotic kinesin-like proteins 1 and 2 (MKLP1 and MKLP2). We first evaluated and compared the localization of these markers in early Telophase I, the stage in which MBs form. We used line scans to quantify localization across the MB. PRC1 (MB core marker) was enriched in two disc-like structures flanking and associated with the dark zone where microtubule signals were absent[3] (Fig. 1c, d), whereas MKLP2 (MB arms marker) colocalized more broadly with microtubules at midzone spindles (Fig. 1e, f). These localizations are similar to mitotic MBs[3]. Both proteins sometimes had concave staining patterns on the PB side of the dark zone, tracking with and ending at the "socket" shape of the microtubules on the PB side. Centralspindlin component[14] MKLP1 (MB ring marker) localization was distinct because we also observed a bulging, cap-like structure (cap) that surrounded the microtubules on the egg side and always protruded towards the forming PB, going beyond the socket-shaped microtubule ends (Fig. 1g–i). These localization patterns were not Telophase I-specific because we also observed the same localization patterns in Telophase II (Supplementary Fig. 1). Finally, we also examined localization of MKLP1, PRC1, and MKLP2 at different meiotic stages from Metaphase I through Metaphase II and observed dynamic localizations largely similar to mitotic cytokinesis[3]. Specifically, MKLP1 localized to the spindle midzone at Anaphase I and the mMB at Telophase I and Metaphase II (Supplementary Fig. 2a). PRC1 localized to microtubule tips at Metaphases I and II, and to the spindle midzone at Anaphase I (Supplementary Fig. 2b). MKLP2 localized to the spindle at Metaphases I and II and the spindle midzone at Anaphase I (Supplementary Fig. 2c). Therefore, mMB proteins have dynamic localization during different cell-cycle stages of meiosis I. Of note, MKLP1 and PRC1 remained with apparent mMBRs after cytokinesis completion (Metaphase II), suggesting that the mMB components are recruited de novo during cytokinesis of meiosis II (Supplementary Fig. 1).

We further evaluated mMB cap genesis by tracking the localization of exogenously expressed *Mklp1-Gfp* in live oocytes (Supplementary Movie 1). Consistent with localization of endogenous MKLP1 in fixed oocytes (Supplementary Fig. 2a), MKLP1-GFP enrichment was observed upon Anaphase I onset (Fig. 1j, timepoint 00:00) along the midzone spindle. The mMB cap was fully formed by early Telophase I (Fig. 1j, timepoint 01:00 and inset). By late Telophase I the mMB further matured and the cap structure disappeared (Fig. 1j, timepoint 02:20 and inset). Once the Metaphase II spindle formed (Fig. 1j, timepoint 03:00), MKLP1-GFP localized between the egg and PB with no further

observable changes in its morphology. In subsequent figures, Anaphase I, early Telophase I, and late Telophase I (all pre-abscission stages (Supplementary Fig. 2)) are defined by the localization of MKLP1 and morphologies of the mMB cap and midzone spindle. We employed this classification because oocytes do not undergo Anaphase I onset synchronously and it therefore allows for precise cell cycle-stage comparisons.

To determine if the cap structure is also observed with other MB ring markers or if this structure is unique to MKLP1, we probed early Telophase I-stage oocytes for additional markers commonly used for mitotic MB ring identification: Rac GTPase-activating protein 1 (RAC-GAP1), also a Centralspindlin component[14]; Citron Kinase (CIT; more commonly called CITK), a midbody kinase[15]; epithelial cell transforming 2 (ECT2), a guanine nucleotide exchange factor[16]; and centrosomal protein 55 (CEP55), a midbody structure regulator[17]. The images revealed that RACGAP1 localized like MKLP1 (ring + cap), whereas CITK, ECT2, and CEP55 localized only at the ring, and we did not detect them in the cap (Supplementary Fig. 3a). We then evaluated RACGAP1 and CITK colocalization with MKLP1 by using super-resolution STED microscopy and compared the Manders coefficients, a measure of overlap between pixels (Supplementary Fig. 3b–d). The analyses indicated that there was greater colocalization of pixel signals between MKLP1 and RACGAP1 than between MKLP1 and CITK, which was expected based on their different observed localizations. Thus, these results indicate that the mMB cap consists of at least the Centralspindlin complex proteins, MKLP1 and RACGAP1, and not the other ring proteins tested. These data also demonstrate that mMBs have conserved arm and core structures as mitotic MBs, but oocytes have a modified ring that contains an additional sub-structure that bulges toward the PB in early Telophase I and consists of at least the Centralspindlin complex. We refer to this sub-structure here as the mMB cap (Fig. 1i).

### Meiotic midbody remnant formation

In somatic cells, the final stage of cytokinesis is abscission, where the severing of the microtubule arms and membrane scission occurs. Abscission leads to the extracellular release of the membrane-bound MB remnant (MBR)[10]. After release, the MBR can be internalized, after which it is then called a MBsome[9,10]. Meiotic MBR (mMBR) formation and abscission have not yet been evaluated in mouse oocytes. To determine if mMBRs form, we first probed Metaphase II-arrested eggs with anti-MKLP1. The images showed that the mMB left the egg and the resulting mMBR was sandwiched between the egg and the PB, bound by their membranes that were marked by phalloidin-based actin staining (Fig. 2a). To confirm the fate of mMBs during and after cytokinesis, we live-cell imaged oocytes exogenously expressing *Mklp1-mCherry* as the mMB and mMBR marker[7,18,19], and *Gap43-eGfp* as the membrane marker[20] (Supplementary Movie 2). Consistent with mMBR formation observed in fixed eggs, we first observed MKLP1-mCherry signal surrounded by the membrane marker after Anaphase I onset (Fig. 2b, timepoints 00:20 through 01:40). MKLP1-mCherry signal remained in a distinct space between the egg and the PB after late Telophase I (Fig. 2b, 02:00 onwards), suggesting that abscission occurs. To further evaluate mMBR formation, we next examined the recruitment of one of the endosomal sorting complexes required for transport-III (ESCRT-III) effector proteins, charged multivesicular body protein 4B (CHMP4B), at late Telophase I[21,22]. ESCRT III proteins colocalize with sites of microtubule constriction where abscission later occurs[23]. One band of CHMP4B immunoreactivity suggests one abscission site, whereas two parallel bands flanking the dark zone suggest two abscission sites[23–26]. We found CHMP4B initially recruited as a single band during early Telophase I (Fig. 2c, top row), and then localized to both sides of the mMB arms and flanked the dark zone during late Telophase I (Fig. 2c, middle row). Of note, we found persisting mMB arms and recruited CHMP4B at Metaphase II, suggesting

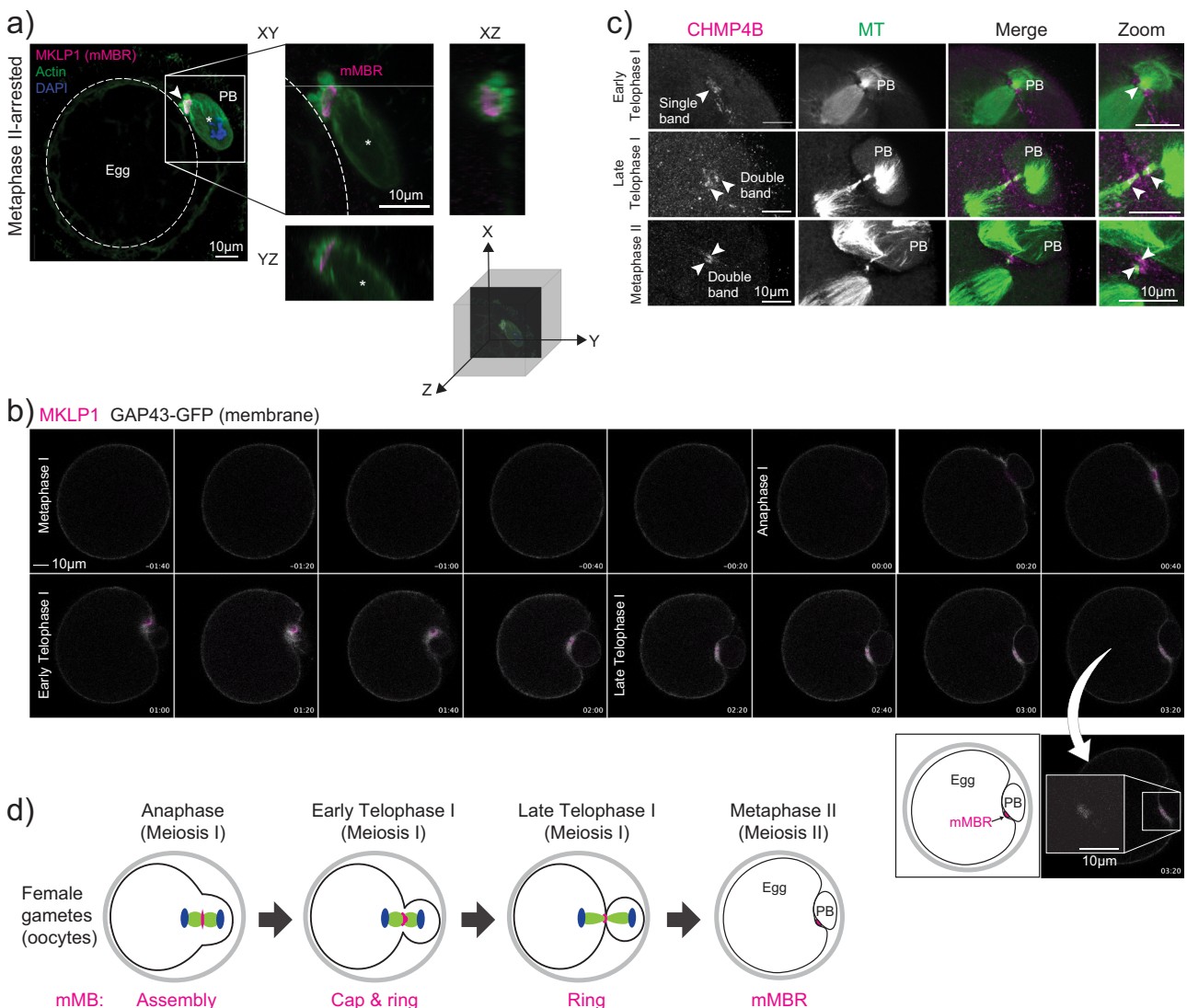

**Fig. 2 | Meiotic midbody remnant formation. a** Representative confocal images of MKLP1 (magenta) and cell boundaries (actin, green) at Metaphase II. On the left, whole egg image with membrane delineated with dotted, white circle is shown. Square indicates the region shown in the zoom on the right side with XY, YZ, and XZ views with a three-dimensional coordinate system with axes depicting orientation of the different views. Asterisk marks the polar body. **b** Representative still images from confocal live-cell imaging of oocytes undergoing cytokinesis, expressing MKLP1-mCherry (magenta) and GAP43-GFP (gray). Stages of meiosis are labeled.

Timepoint 00:00 represents the beginning of Anaphase I onset. Below is a zoom of the mMBR at Metaphase II and a cartoon interpretation. **c** Representative confocal images showing localization of CHMP4B (magenta) relative to microtubules (green; tubulin) and to the MB region in early Telophase I (top panels), Late Telophase I (middle panels), and Metaphase II (bottom panels). Arrowheads indicate bands of CHMP4B enrichment. All experiments were conducted three times. **d** Schematic summarizing the timing of formation mMB cap, its regression and mMBR formation.

that abscission does not take place until after the establishment of the Metaphase II spindle (Fig. 2c, bottom row), consistent with previous reports in mitotic cells that describe abscission taking place after daughter cells enter G1[27]. Although we did not examine membrane scission and further experimentation is required to confidently establish the number of abscission sites, the data suggest that mMBs are abscised from the egg into mMBRs. Together, the data indicates that a cap-containing mMB forms in early Telophase I, the cap resolves in late Telophase I, and an mMBR forms in Metaphase II (Fig. 2d).

## Microtubules are associated with mMB cap formation

Because the cap sub-structure caught our attention, we next sought to understand what drives its formation. One of the major observable differences during cytokinesis between oocytes and most other mammalian somatic cells is that oocytes undergo an asymmetric division forming a large egg and small PB[11,28] (Fig. 1a). Because of this difference, we hypothesized that the asymmetric division plays a role

in mMB cap formation. To address our hypothesis, we induced symmetric division by gently compressing oocytes at Metaphase I. This method results in two daughter cells of equal size that are viable but have reduced developmental competence[29]. When oocytes were forced to undergo symmetric division, we observed loss of the ball/ socket shape of the midzone spindle as microtubules appeared similar on either side of the midzone. Importantly, the mMB cap disappeared (Fig. 3a and Supplementary Movie 3), suggesting that mMB cap formation requires asymmetric cytokinesis. Alternatively, the cap may be sensitive to changes in pressure and other mechanical perturbations induced by compression, possibilities that could be further evaluated.

Two major cellular components of cell division are the midzone spindle microtubules and the actomyosin ring[30]. Because both the mMB cap and ball/socket-like structure of the midzone spindle were absent when oocytes underwent symmetric division, we hypothesized that microtubules are required for cap formation. To test this hypothesis, we perturbed microtubules during mMB formation (early

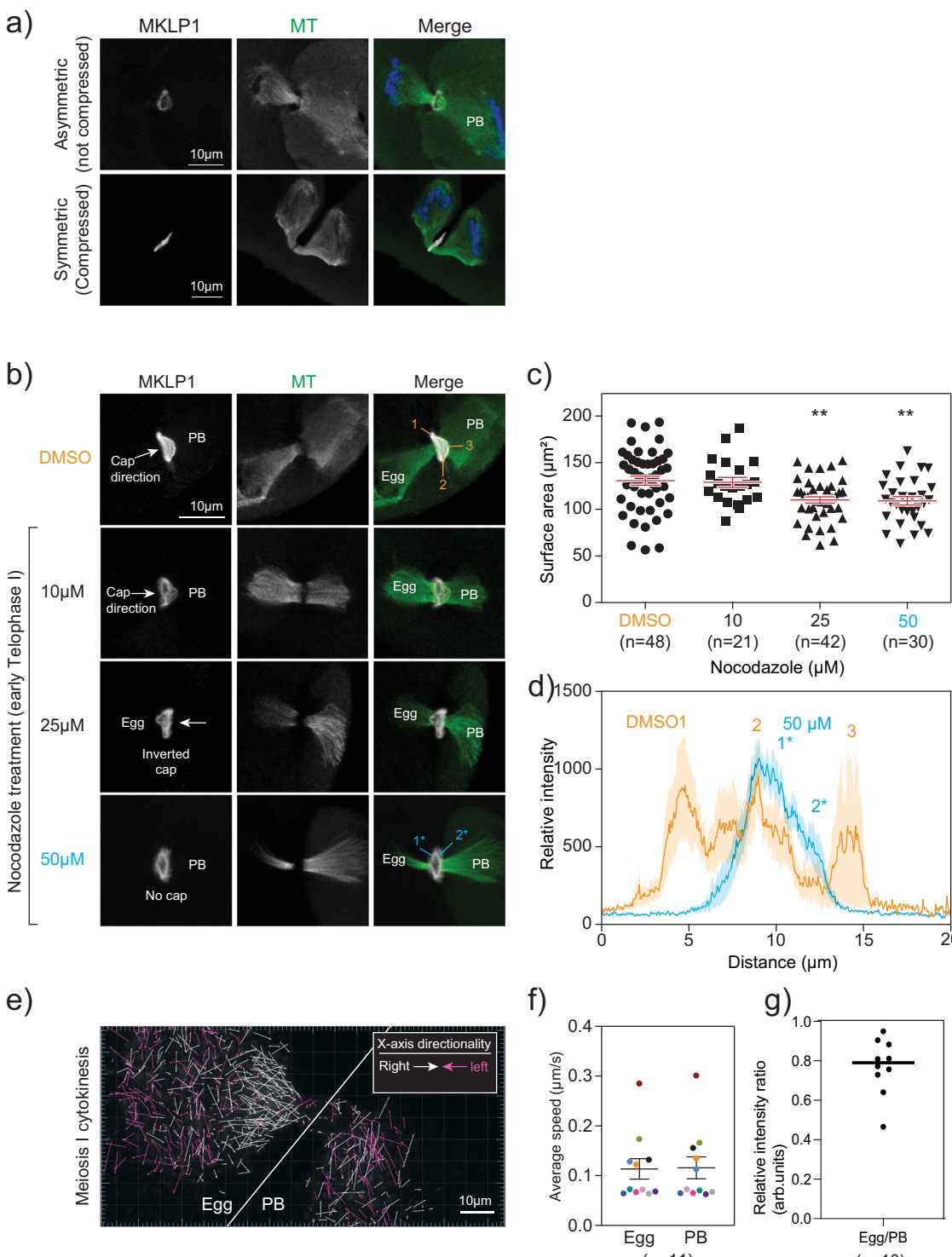

**Fig. 3 | Microtubules are required to form the meiotic midbody cap.**
**a** Comparison of the mMB cap structure (MKLP1, gray) and microtubules (green; tubulin) when oocytes undergo asymmetric (top panels) or symmetric (bottom panels) divisions. PB = polar body **b** Representative confocal images of nocodazole dose-dependent distortion and reduction in the mMB cap (MKLP1; gray) and midzone spindle (tubulin; green). The regions of MKLP1 numbered are the peaks detected in line scans in (**d**) and the arrows indicate the direction of the cap bulge. **c** Quantification of surface area occupied by ring after nocodazole treatment. One-way ANOVA; 3 replicates; number of oocytes in DMSO: 48, 10 μM: 21, 25 μM: 42, 50 μM: 30; **$p = 0.0014$ (25 μM) and 0.0026 (50 μM). **d** Line scan plots of MKLP1 intensity from control (orange) and nocodazole-treated (blue) oocytes in (**b**). Shading reflects the standard error of the mean. The numbers reflect the numbers labeled in the corresponding images. **e–g** Comparison of microtubule polymerization and dynamics during early telophase I by confocal live-cell imaging of oocytes expressing EB3-GFP. **e** Representative still image from live-cell confocal imaging of oocytes undergoing cytokinesis and expressing EB3-GFP. X-axis directionality of arrows towards left (pink) and right (white) are colored coded in (**e**). Single z-plane was selected based on clear visualization of the dark zone in oocytes with the midzone spindle oriented parallel to the imaging plane. White line delineates the egg (left) and PB (right) sides. **f** Average speed of EB3-GFP puncta in egg versus PB; 11 oocytes. Data points collected from the same dividing oocyte are color-matched. g) Relative EB3-GFP intensity ratio in PB compared to egg. Unpaired Student's t-test, two-tailed; 10 oocytes; ** $p < 0.01$, **** $p < 0.0001$. Data are presented as mean values +/- SEM. arb. units = arbitrary units.

Telophase I) using nocodazole treatment and found that increasing concentrations of this microtubule depolymerizer caused deformation of the cap at lower concentrations and complete cap regression at 50 μM, the highest concentration tested (Fig. 3b, c). Analysis of MKLP1 pixel intensity across the midbody bridge in DMSO controls, showed that there were three peaks of MKLP1 staining. These peaks corresponded to the two sides of the ring and the cap. Upon 50 μM nocodazole treatment, only two peaks of MKLP1 staining were apparent, corresponding to the sides of the ring (Fig. 3d) and a cap peak was not detected. Notably, at these doses, microtubules were still present, but the ball/socket morphology of the microtubules disappeared. We also observed incidences of inverted mMB caps pointing toward the egg (Fig. 3b, 25 μM), which we suspect originated from a change in microtubule dynamics and/or abundance upon nocodazole treatment.

We next evaluated microtubules in the mMB by live-cell imaging mMB formation in oocytes expressing *Eb3-eGfp* (end-binding protein 3), a marker of a plus-end microtubules often used as an indicator of microtubule dynamics[31] (Fig. 3e and Supplementary Movie 4). By comparing microtubule polymerization speed and density between the egg and PB sides, we found that although microtubule polymerization speeds were similar (Fig. 3f), microtubules were more abundant on the egg side (Fig. 3e, g). Furthermore, many egg-side EB3 movements were directed towards the mMB whereas PB-side EB3 movements were random in directions (Fig. 3e; white arrows). These observations correlated with the directionality of the cap and the ball/socket morphology of the midzone spindle. We also tested a possible requirement of actin in mMB cap formation. After treatment with latrunculin A (Lat A), a pharmacological agent that depolymerizes actin, the cap disappeared with both 5 μM and 10 μM doses, while the mMB ring remained in the control DMSO group (Supplementary Fig. 4a, c). We confirmed actin depolymerization by detecting actin with phalloidin staining: we observed organized actin at the egg and PB membranes in controls (Supplementary Fig. 4b, orange arrows in top row), and diffused and disorganized actin signal with 5 μM and 10 μM Lat A treatments. But, because disruption of actin also perturbed spindle microtubules, we could not conclude that actin has a direct role in mMB cap formation. Notably, MKLP1 localized to the ring in the absence of either microtubules or actin. From the compression, nocodazole, and EB3 results, the data suggest that microtubules are required to form the cap structure.

## Meiotic midbodies are enriched in ribonucleoproteins

Studies on MB functions have extended beyond regulatory functions of cytokinesis, and now indicate their signaling capabilities[9] and ribonucleoprotein (RNP) properties[13,32–34]. An array of proteins involved in translation, translational regulation, and RNA molecules are enriched in mitotic MBs[33,34,35]. The enrichment of these components suggests translational capabilities within MBs and offers an explanation as to how its inheritance after abscission as an MBR could regulate cellular function in a cell type-specific manner. These properties are unknown in oocytes. Therefore, we investigated whether the mMB of early Telophase I oocytes (pre-abscission) have RNP characteristics, by assessing: 1) enrichment of RNA molecules, 2) increased localization of translation machinery, and 3) localized active translation. By performing fluorescence in situ hybridization (FISH) to detect the polyadenylated (Poly-A) tail of transcripts, we found enrichment of Poly-A signal in mMBs over the background signal of mRNAs in the egg cytoplasm (Fig. 4a). By immunocytochemistry, we also observed enrichment of small (RPS3, RPS6, and RPS14) and large (RPL24) ribosomal subunit proteins in mMBs (Fig. 4b). Finally, to detect nascent, active translation in mMBs, we carried out a Click-chemistry-based assay that detects *L*-homopropargylglycine (HPG), a methionine-analog, that is integrated into newly translated proteins during acute incubation[36]. Similar to mRNAs and ribosome subunit proteins, we found enrichment of nascent translation in oocyte early Telophase I

mMBs (Fig. 4c). We confirmed the specificity of the HPG signal when we observed its decrease after treating oocytes with cycloheximide and puromycin, two translation inhibitors, and observed ~40% reduction in HPG signal (Fig. 4c–e). We note that the HPG signal did not completely disappear. It is possible that the timing of adding the inhibitors and the time it takes for translation to shut down allows for some translation to occur. Alternatively, it may be difficult for chemicals to penetrate this protein-dense region as is suggested by the nocodazole experiments that did not completely depolymerize microtubules (Fig. 3b). These findings that there are mRNA, both large and small ribosomal subunit enrichment, and active translation support the model that mMBs have RNP properties, similar to mitotic MBs and MBRs[12].

## mMB cap demarks translation boundary between oocyte and PB

One feature of the HPG/translation signal in mMBs was that its localization was similar to cap localization (Figs. 1g, 4c and Supplementary Fig. 3a). To further evaluate the relationship between the cap and the translation signal, we imaged early Telophase I-staged oocytes (pre-abscission) to detect MKLP1 and HPG Click-IT. The HPG translation signal was enriched on the egg side of the cap, and absent on the PB side (Fig. 5a, and Supplementary Movie 5). This observation led to the hypothesis that the cap keeps RNAs and/or proteins synthesized at mMBs in the egg and prevents them from going into PBs. Because we observed that nocodazole treatment disturbed the mMB cap (Fig. 3b–d), we compared nascent translation in mMBs with an intact cap to translation when the cap was disrupted by nocodazole treatment. In contrast to control oocytes, in which HPG translation signal stopped at the MKLP1 cap signal, in oocytes with a disrupted cap, we saw two differences: 1) the translation signal no longer filled the mMB space bounded by the cap and appeared disorganized, and 2) there was HPG signal leakage into the PB that appeared as streaks (Fig. 5b, c and Supplementary Movie 6). These results suggest that the mMB cap encapsulates translation activity and products and is the boundary for nascent translation between eggs and PBs in early Telophase I (Fig. 5d).

## mMB cap is required for developmental competence

To test the model that the cap prevents nascent translation (marked by HPG) from leaving the egg and that it is important for downstream developmental competence, we used laser ablation to puncture the cap. Under brightfield illumination, the mMB was easily detectable because of its distinctive refraction (Supplementary Fig. 5a). In early Telophase I oocytes, a time point before abscission occurs, we employed a multi-photon laser ablation (780 nm wavelength) to partially disrupt mMB cap integrity. Additional control groups were included, in which oocytes were exposed to the same ablation protocol at either the egg side or the PB side of the spindle (Fig. 6a and Supplementary Fig. 5a, c). All ablated oocytes successfully extruded PBs (Supplementary Fig. 5b). We confirmed that ablation disrupted the mMB cap integrity by detecting MKLP1 in control and cap-ablated oocytes. Control-ablated oocytes had intact mMB rings and caps, whereas the cap-ablated oocytes had only intact MKLP1 rings and had a hole in the anti-MKLP1 labeled cap (Fig. 6a and Supplementary Fig. 5c).

We next determined whether disrupting mMB cap integrity resulted in leakage of newly translated proteins into PBs. Early Telophase I oocytes were either not exposed to laser ablation (non-ablated controls) or exposed to a multi-photon laser ablation in the cytoplasm (cytoplasmic ablation), at the egg side of the spindle (egg MT-ablated) or at the mMB cap (to disrupt its integrity); we excluded the PB-side MT ablation in this experiment. After ablation, we assessed the localization of nascent proteins using Click-IT HPG labeling. In controls, HPG signals were limited to the mMB cap area and bounded specifically at the egg side of the cap (Fig. 6b, c). Importantly, disrupting the mMB cap integrity resulted in HPG signal leakage which appeared as streaks that extended beyond the boundaries of the mMB cap (Fig. 6b,

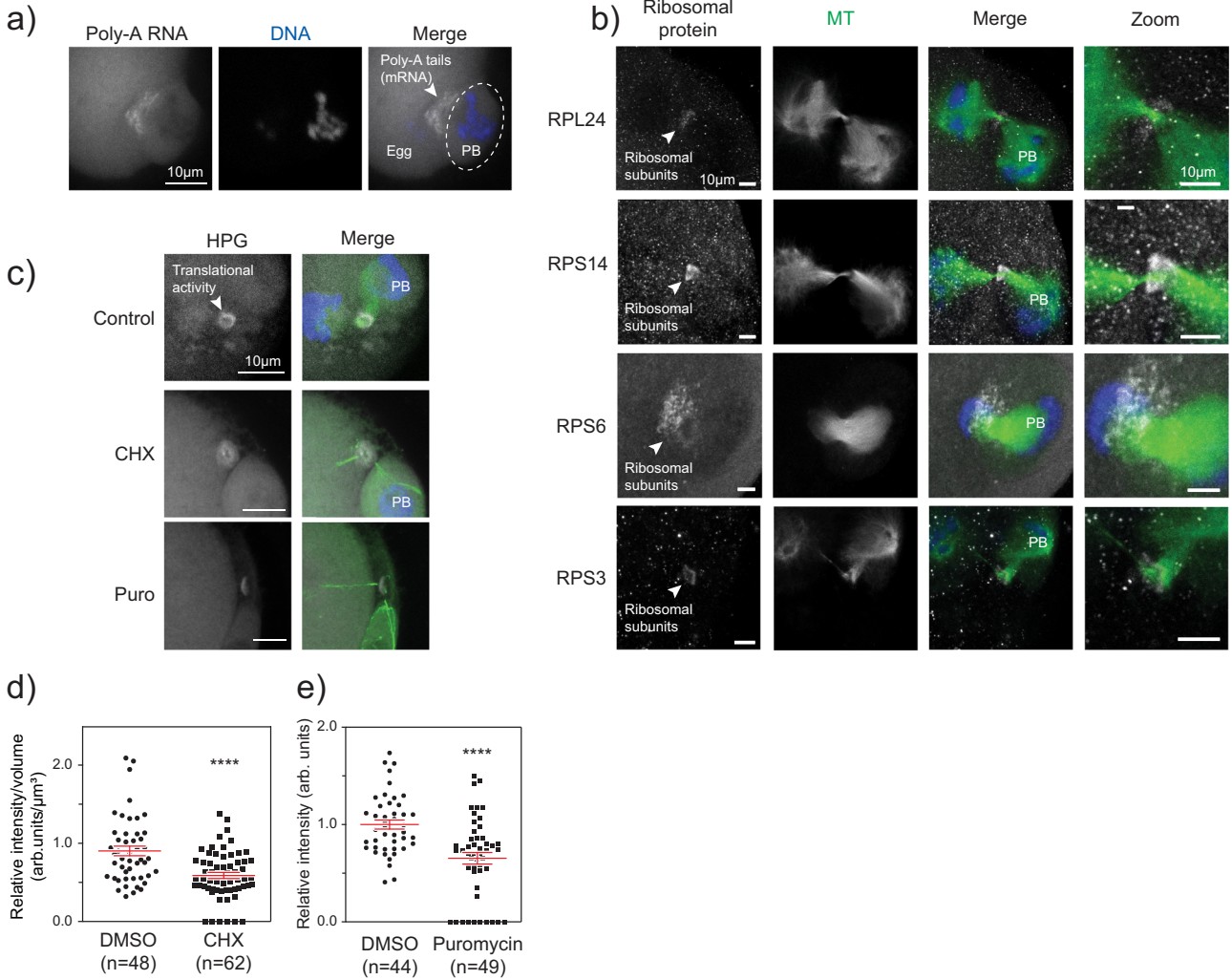

**Fig. 4 | The meiotic midbody is translationally active. a** Confocal images showing localization of polyadenylated (Poly-A) tails of mRNA molecules at mMBs, detected by RNA FISH. The polar body (PB) is encircled; the white arrowhead highlights the FISH signal enriched in the midbody region. This experiment was replicated 3 times. **b** Confocal images representing localization of small (RPS3, RPS6, and RPS14) and large (RPL24) ribosomal subunits (gray) at the midbody ring region relative to midzone spindle (green; tubulin). The white arrowhead points to the ribosomal subunit proteins. These experiments were repeated 3 times. **c** Representative confocal images of translational activity (gray; HPG) by Click-IT

assay with and without inhibition of translation with cycloheximide (CHX) or puromycin (Puro); midzone spindle (green; tubulin). HPG homopropargylglycine. **d** Quantification of translation signal at the midbody region in control versus CHX-treated cells. Unpaired Student's t-test, two-tailed; 3 replicates. number of oocytes in DMSO: 48, CHX: 62. **e** Quantification of translation signal at the midbody region in control versus puromycin-treated cells. Unpaired Student's t-test, two-tailed; 2 replicates. number of oocytes in DMSO: 44, Puro: 49. ****$p < 0.0001$. Scale bars = 5 μm (zooms). Data are presented as mean values +/- SEM; arb. units = arbitrary units.

c). The percentage of oocytes harboring leaked HPG streaks was significantly higher in mMB cap-ablated oocytes when compared to control-ablated and non-ablated oocytes (Fig. 6b, c). These data and the nocodazole treatment data (Fig. 5b, c) support the model that the cap functions to keep mMB-translated proteins in the egg.

To determine if the mMB-localized proteins are required for developmental competence of the egg, we parthenogenetically activated the non-ablated, control egg MT-ablated, and mMB cap-ablated Metaphase II-eggs with strontium chloride and cultured the resulting parthenotes (i.e. embryos made without sperm) for two days. We chose to activate eggs as a proxy for fertilization because the timing of ablation procedures and the processing steps required for in vitro fertilization were not experimentally compatible. Approximately 80% of egg MT-ablated parthenotes activated and cleaved to the two- or four-cell embryonic stages. In contrast, only ~25% of cap-ablated parthenotes developed past the egg/one-cell stage (Fig. 6d) despite completion of meiosis II as visualized by 2nd polar body bulging. Similar developmental defects were observed in the mMB cap-ablated group when compared to parthenotes derived from cytoplasm-

ablated oocytes (Supplementary Fig. 6a). These data support the model that the mMB cap is required to retain mMB translation products within the egg which later support developmental competence and preimplantation embryo development (Fig. 6e).

Our data identify mMBs in mouse oocytes, showing that they have a specialized cap structure, have RNP enriched structures, and are required for developmental competence. We find that the cap structure contains, at minimum, the Centralspindilin complex proteins MKLP1 and RACGAP1, the presence of which could explain how the cap forms. Centralspindilin is a heterotetrameric complex that bundles microtubules in the cytokinesis furrow[37–39]. Crystal structures of Centralspindilin from *C. elegans* (called ZEN-4/CYK-4) indicate that the complex has biochemical properties that can phase separate[39]. Although we did not explore phase separation here, it is possible that its propensity to phase separate could also be involved in cap formation.

Open microtubule bridges containing MBs exist in diverse systems and allow for sharing of materials between dividing cells. For example, in *Drosophila* gonads, germ cell cysts develop from

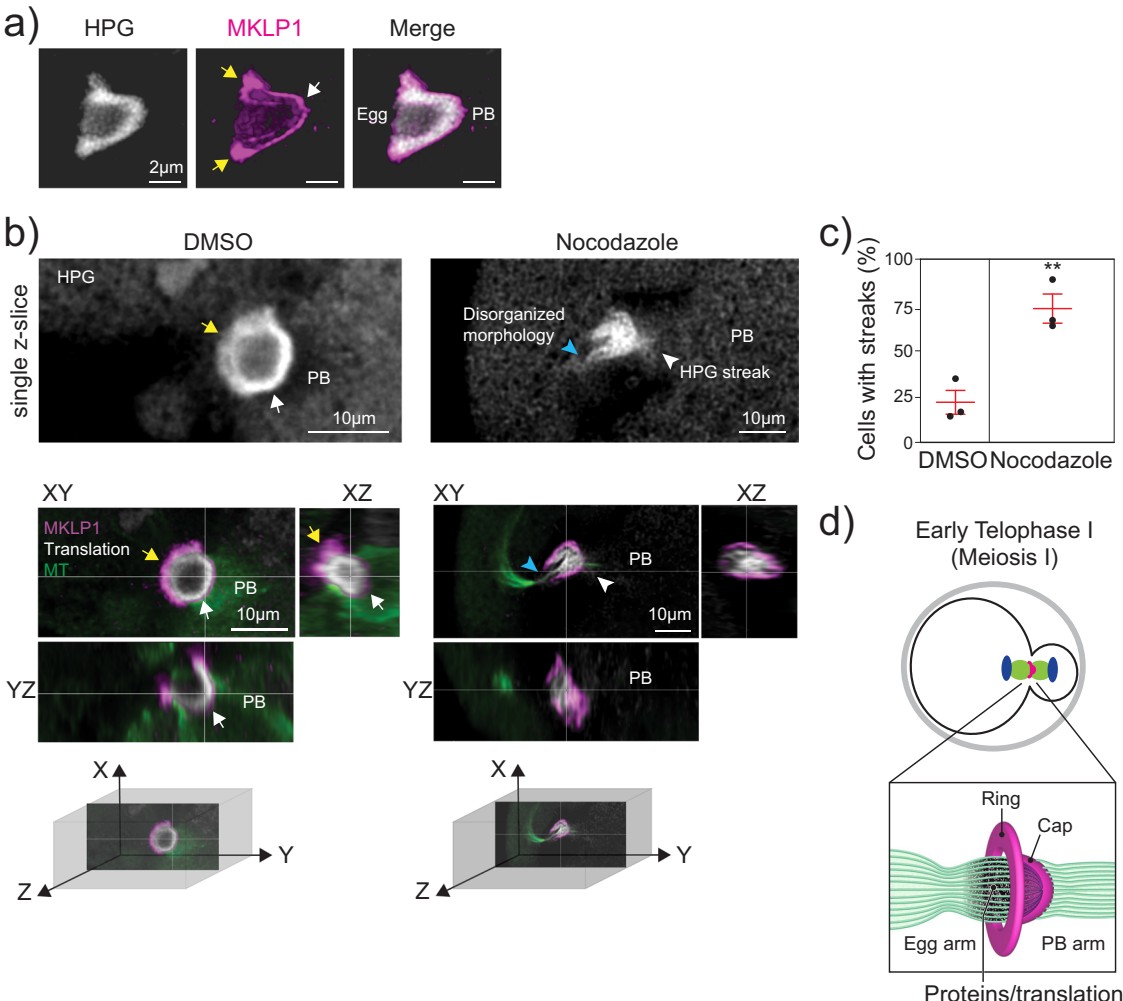

**Fig. 5 | The meiotic midbody cap bounds translation. a** Representative 3D reconstruction of confocal image showing translation localization (gray) labeled with HPG relative to MKLP1 cap (magenta). Arrows point to boundaries in the egg (yellow) and at the polar body (PB) (white). **b** Representative confocal images showing localization of mMB marker MKLP1 (magenta) in relation to translation signal from Click-IT labeling (gray) with views at XY, XZ and YZ planes. The panels on the left are control, DMSO-treated oocytes (intact cap). mMB ring (yellow arrows) and mMB cap (white arrow) are highlighted. The panels on the right are nocodazole-treated oocytes (disrupted cap). The blue arrow indicates disrupted mMB space. Next to confocal images are three-dimensional coordinate systems with axes depicting orientation of the different views. Above the colored confocal images are zooms of a single plane of HPG signal with white arrows pointing to HPG. **c** Proportion of oocytes with HPG leakage in streaks. Unpaired Student's t-test, two-tailed; 3 replicates; number of oocytes in DMSO: 42, Nocodazole: 44. **$p = 0.0076$; Data are presented as mean values +/- SEM. **d** Schematic summarizing the finding that active translation occurs in the mMB region and is bounded by the mMB cap in early Telophase I.

incomplete cytokinesis. The result of incomplete cytokinesis is the formation of open intracellular bridges that allow sharing of molecules and organelles, a process essential for oocyte and spermatocyte development[40–42]. A similar mechanism exists in mouse and human testes, where mitotically dividing spermatogonia undergo asymmetric cytokinesis[43,44] and form intracellular bridges, leading to syncytia formation[45]. A key protein involved in forming these bridges is TEX14. *Tex14* knockout male mice are infertile because spermatocytes cannot complete meiosis[46]. In the fetal mouse ovary, germ cells are also connected by intracellular bridges which later break down after birth[47]. Mouse preimplantation embryos also have persistent microtubule bridges between blastomeres (the individual cells of an embryo), and it is speculated that these provide a mechanism for controlling cell division while the embryo begins to polarize[48]. In contrast, our data suggest that a mMB cap is like a gate that closes what could otherwise be a leaky intracellular bridge to keep essential materials in the egg during early Telophase I. In addition, our findings that mMBRs form, suggest a mechanism that could later affect cell fate in embryos. After meiosis, another wave of asymmetric divisions occurs as

preimplantation embryos acquire 32 cells. Recent work demonstrated that RNAs during this time are asymmetrically enriched at basal regions of the outer blastomeres and their movement depends upon microtubules[49]. We speculate that mMBR released from eggs can later act as signaling organelles during fertilization or pre-implantation embryogenesis if they are phagocytosed by the developing embryo. Alternatively, the mMBR may harbor maternal molecules that are inhibitory to early embryogenesis and therefore need to be sequestered away. Further insight into the identity of the RNA transcripts and proteins in mMBRs is needed to understand their potential roles in embryo development.

The cytoplasm of mammalian eggs sustains meiotic divisions and early embryonic development with a fixed pool of maternal transcripts that are activated and translated in a regulated fashion[50–52]. At the same time, the changes oocytes undergo throughout meiosis happen in a single cell cycle, emphasizing the need for oocytes to optimize and regulate the protein synthesis process. Spatiotemporal control of translation is found across forms of life as an energy-efficient means to meet different needs during cell cycle and throughout different

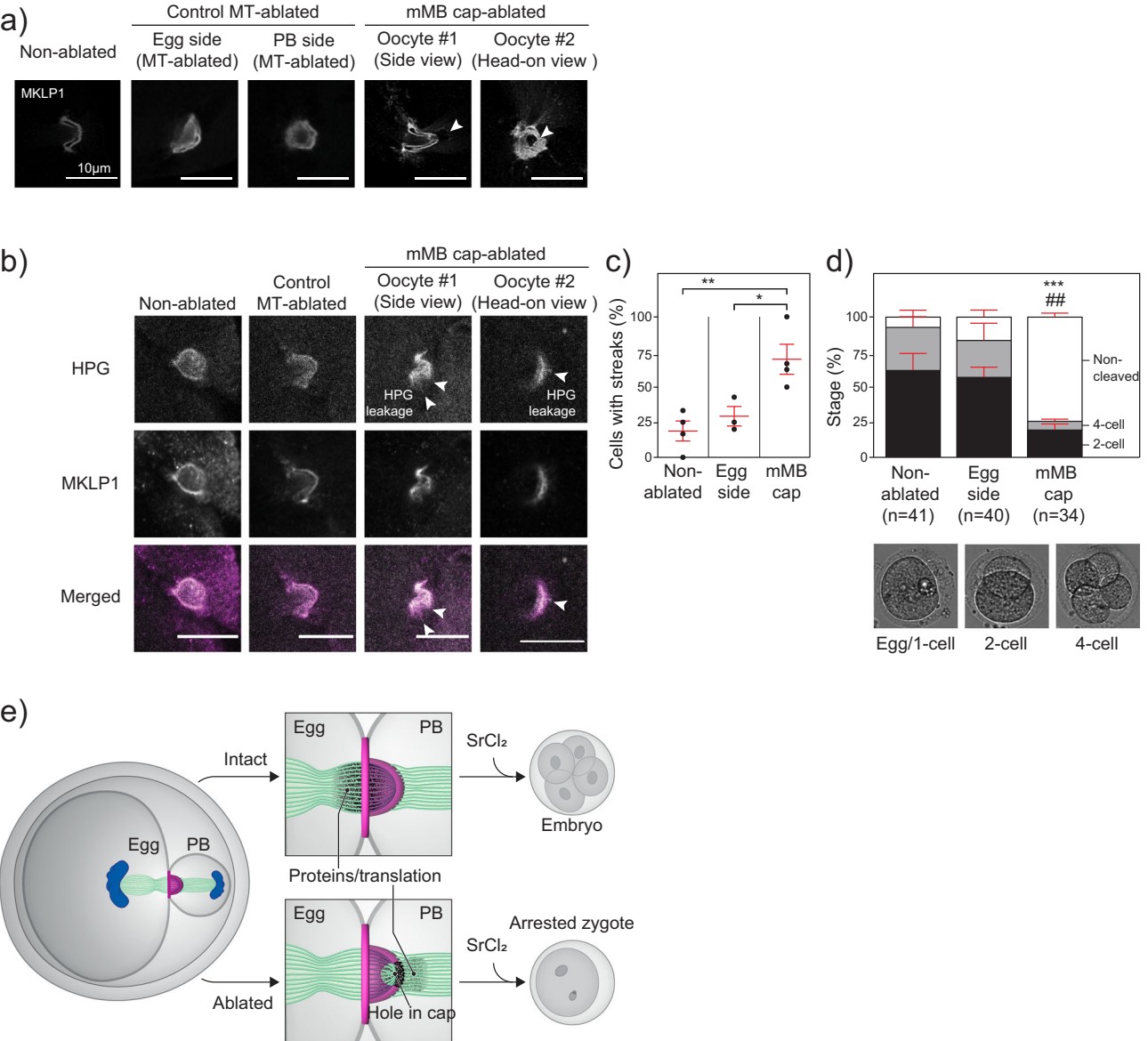

**Fig. 6 | The meiotic midbody cap is required for developmental competence.**
**a** Representative confocal images (single z-plane) of mMB caps (MKLP1) of oocytes from non-ablated, control MT-ablated (egg side and polar body (PB) side), and mMB cap-ablated oocytes. Two mMB cap-ablated oocytes are shown, one with a side view (oocyte #1) and one with a head-on view (oocyte #2). Arrowheads indicate where laser ablation took place. **b** Representative confocal images (single z-plane) of nascent translation (HPG; gray) and MKLP1 (magenta) in non-ablated, control egg side MT-ablated, and two mMB cap-ablated oocytes (oocyte #1 and oocyte #2). Arrowheads indicate where nascent translation leakage was observed by HPG signal in the form of streaks. **c** Quantification of the percentage of oocytes showing HPG streaks. One-way ANOVA Tukey's multiple comparisons test; two-sided; 4 replicates

for non-ablated and mMB cap-ablated, and 3 replicates for MT-ablated; number of oocytes in non-ablated: 15, egg MT-ablated: 16, mMB cap-ablated: 27. $*p = 0.0325$; $**p = 0.0066$. **d** Images of parthenotes after ablation, activation and development in vitro. The graph above the images quantifies the percentage of parthenotes at each developmental stage. $***p = 0.0005$ mMB cap-ablated, non-cleaved cells compared to non-ablated group; $##p = 0.0014$ mMB cap-ablated, non-cleaved cells compared to the control-ablated group. Two-way ANOVA Tukey's multiple comparisons test; two-sided; 2 replicates; number of parthenotes in non-ablated: 41, egg MT-ablated: 40, mMB cap-ablated: 34. Scale bars = 10 μm. Data are presented as mean values +/- SEM. **e** Model summarizing a requirement for meiotic midbody caps in mouse oocytes based on ablation experiments. $SrCl_2$, strontium chloride.

regions of a cell[53,54]. Oocytes lack both an interphase and an S-phase between meiosis I and II, and they are transcriptionally silent until zygotic genome activation in embryos at the 2-cell stage in mice and 8-cell stage in humans. Because eggs from mMB cap-ablated groups did not develop efficiently upon activation, we hypothesize that the mMB cap ensures nascent proteins remain in the egg, a function critical to subsequent embryonic development (Fig. 6d). Our findings describe a mechanism by which oocytes ensure their quality and developmental competence in preparation for supporting embryogenesis. Thus, we propose a model in which the mMB cap is an evolutionary adaptation of the microtubule bridge in oocytes to

ensure the developmental competence of eggs after fertilization by acting as both a translation hub and as a barrier that retains maternally derived proteins in the egg (Fig. 6e). Therefore, future studies assessing mMBR fate and identification of mMB and mMBR proteins will be critical for understanding how embryos may benefit from mMBR formation and possible inheritance.

## Methods
### Oocyte and egg collection and culture
Sexually mature CF-1 female mice (*Mus musculus*) (6–10 weeks of age) were used for all experiments (Envigo, Indianapolis, IN, USA). Only

females were used because we only evaluated oocytes in this study. All animals were maintained in accordance with the guidelines and policies from the Institutional Animal Use and Care Committee at Rutgers University (Protocol# 201702497) and the Animal Care Quality Assurance at the University of Missouri (Reference# 9695). Experimental procedures involving animals were approved by these regulatory bodies. Mice were housed in a room programmed for a 12-hour dark/light cycle and constant temperature (between 70–74°F), humidity (50%) and with food and water provided ad libitum. Prior to each experiment, approximately 3–5 females were injected intraperitoneally with 5 I.U. of pregnant mare serum gonadotropin 48 h prior to oocyte collection (Lee Biosolutions, Cat# 493-10). Prophase I-arrested oocytes were harvested[55]. Briefly, cells were collected in minimal essential medium (MEM) containing 2.5 µM milrinone (Sigma-Aldrich, M4659) to prevent meiotic resumption, and cultured in Chatot, Ziomek, and Bavister (CZB) media[56] without milrinone in a humidified incubator programmed to 5% $CO_2$ and 37° C for 11–12 h for cytokinesis at meiosis I, or overnight for certain drug treatments.

For evaluating midbodies in meiosis II, ovulated eggs were activated with 10 mM strontium chloride (Sigma Aldrich, Cat# 25521) to induce Anaphase II onset. To collect ovulated eggs, mice were injected with human chorionic gonadotropin (hCG) (Sigma Aldrich, Cat# CG5) 48 h after PMSG injection to stimulate ovulation of Metaphase II-arrested eggs. 14–16 h following hCG injection, eggs were harvested from the ampulla region of the oviducts in MEM containing 3 mg/ml of hyaluronidase (Sigma Aldrich, Cat# H3506) to aid detachment of cumulus cells. Eggs were then transferred to center-well organ culture dish (Becton Dickinson, Cat# 353037) with activation media, consisting of $Ca^{2+}$/$Mg^{2+}$-free CZB with 10 mM of strontium chloride, and cultured in a humidified incubator programmed to 5% $CO_2$ and 37° C. After 3 h, activated eggs were cultured for 3 additional hours in KSOM + amino acids media (Sigma Aldrich, Cat# MR-106-D). For parthenogenetic activation of eggs, the activation and KSOM media were supplemented with 5 µg/ml cytochalasin D (Sigma Aldrich, Cat# C2743). Parthenogenetically activated eggs were incubated for 48 h in KSOM + amino acids media to assess embryo cleavage rate.

For microinjection, collected oocytes were maintained arrested at Prophase I with milrinone before injection to prevent nuclear disruption and after injection to allow translation of cRNAs. To induce symmetric division of oocytes, cells were compressed at Metaphase I[29]. Briefly, after culturing for 8 h (Metaphase I time point), cells were transferred to a 5–7 µl drop of CZB covered with mineral oil (Sigma Aldrich, Cat# M5310). A glass cover slip was placed on top of the media drop and pressed down on the edges to spread the media to cover the entire surface of the cover slip. The cover slip was then pressed down until oocytes flattened and the zona pellucida became indistinguishable from the cell membrane. Cells were then cultured for an additional 3 h to observe cytokinesis.

## Inhibition and disruption of mMB
To depolymerize microtubules and actin during mMB formation in early Telophase I, oocytes were cultured in CZB for 11 h and then transferred to media containing nocodazole (Sigma Aldrich, Cat# M1404) (0, 10, 25, and 50 µM) or latrunculin A (Cayman Chemical Company, Cat# 10010630) (0, 5, and 10 µM) in a center-well dish for 30 additional minutes.

For translation inhibition, oocytes were cultured for 9 h prior to overnight in center-well organ culture dishes with CZB media supplemented with glutamine, containing either cycloheximide at 50 µg/ml (Sigma-Aldrich, Cat# C7698) or puromycin at 1 µg/ml (Sigma-Aldrich, Cat# P7255).

## Ablation of mMB cap by laser ablation
Prophase I-arrested oocytes were cultured in vitro in milrinone-free CZB medium supplemented with 100 nM SiR-tubulin (Cytoskeleton

#NC0958386) in a humidified, microenvironmental chamber (5% $CO_2$ and 37°C) equipped to a Leica TCP SP8 inverted microscope. After culturing cells for 11 h, mMB caps were partially ablated using a multiphoton laser[57]. In brief, a 4 µm² square region of interest within the mMB cap was exposed to a 780 nm wavelength and 60–70 mW power laser beam at the sample plane. For control-ablated oocytes, the cytoplasmic region adjacent to the mMB (cytoplasmic ablated), the egg side of the spindle (egg MT-ablated) or the PB side of the spindle (PB MT-ablated) were exposed to the same protocol. A subset of cap-ablated, control-ablated and non-ablated oocytes were fixed and immunostained with MKLP1 antibody to assess the efficiency of laser ablation and mMB cap disruption.

## Immunofluorescence
Following meiotic maturation, oocytes or activated eggs were fixed in various conditions to detect localization of proteins. For detection of PRC1 (Proteintech, 15617-1-AP, 1:100), CIT-K (BD Biosciences, 611376, 1:100), RACGAP1 (Santa Cruz, sc-271110, 1:50), MKLP1 (Novus Biologicals, NBP2-56923, 1:100), and MKLP2 (Proteintech, 67190-1, 1:100), oocytes were fixed in 2% PFA in phosphate-buffered saline (PBS) for 20 min at room temperature. For detection of ECT2 (Fortis Life Sciences, A302-348A, 1:100), oocytes were fixed in 2%PFA with 0.1% Triton-X in PBS for 20 min at room temperature. For detection of RPS3 (Cell Signaling Technology, 2579 S, 1:30), RPS6 (Santa Cruz, sc-74459, 1:30), RPS14 (Proteintech, 16683-1-AP, 1:30), RPL24 (ThermoFisher, PA5-62450, 1:30), and CEP55 (Proteintech, 23891-1-AP, 1:50), oocytes were fixed in cold methanol (Sigma Aldrich, Cat# A452-4) for 10 min. For detection of CHMP4B (Proteintech, 13683-1-AP, 1:30), zona pellucida were removed from oocytes by brief treatment with acidic Tyrode's solution (Sigma Aldrich, Cat# MR-004-D) and fixed with 2% PFA in PBS for 20 min at room temperature. After fixation, oocytes were incubated in a blocking buffer (0.3% BSA and 0.01% Tween in PBS) for at least 10 min before proceeding. For permeabilization, oocytes were incubated in PBS containing 0.2% Triton-X for 20 min and blocked with a blocking buffer for 10 min. For ECT2, RPS3, RPS6, RPS14, RPL24, CEP55, RACGAP1, and CHMP4B, cells were incubated overnight at 4° C with primary antibody. For all other proteins, primary incubation was performed for 1 h at room temperature. For secondary antibody incubation, oocytes were incubated for 1 h in a dark, humidified chamber at room temperature. Both antibody incubations were followed by three washes in blocking solution, 10 min each. After the last wash, oocytes were mounted in 10 µl of Vectashield (Vector Laboratories, Cat# H-1000) containing 4,6-Diamidino-2-Phenylindole, Dihydrochloride (DAPI) (Life Technologies, Cat# D1306; 1:170) for confocal microscopy.

For super-resolution microscopy using the tau-3X STED module from Leica Biosystems, the same steps as the ones described above for confocal microscopy were followed except for the following changes: 1) antibody concentrations were doubled for primary antibodies and 2) after the third wash following secondary antibody incubation, cells were mounted in 10 µl of EMS glycerol mounting medium with DABCO (EMS, Cat# 17989-10).

## RNA in situ hybridization
To detect RNA molecules, fluorescence in situ hybridization (FISH) against the poly-A tail of transcripts was performed using an oligo-dT probe that consists of 21 thymine nucleotides with a 3′ modification of a fluorophore[58]. Briefly, oocytes were fixed in increasing volumes of methanol-free 4% formaldehyde diluted in RNase-free 1X PBS at 37° C for 45 min. Oocytes were then dehydrated in increasing concentrations of methanol and stored at −20° C until further processing. Oocytes were prepared for hybridization by washing through 1X PBS with 0.1% Tween-20 (PBT), followed by 10% formamide and 2X SSC in nuclease-free water (Wash A). For the hybridization reaction, oocytes were incubated in a 10% formamide, 2X SSC and 10% dextran

sulfate solution in nuclease-free water with 12.5 μM of the probe overnight at 37 °C. After hybridization, samples were rinsed through several volumes of fresh, pre-warmed Wash A and PBT before mounting on 10 μl of Vectashield with DAPI for imaging.

### Nascent protein detection assay

Translation activity at the midbody was assessed by detecting protein synthesis level using an *L*-HPG-translation kit (ThermoFisher, Cat# C10429) as previously described[36]. In summary, oocytes were collected and matured for 11.5 h, then transferred to DMEM medium lacking methionine (ThermoFisher, Cat# 42-360-032) and containing HPG diluted 1:50 for 30–45 min, followed by fixation with 2% PFA in PBS for 20 min at room temperature and subsequent detection of HPG signal by immunofluorescence.

Control and ablated oocytes (mMB cap, egg MT-ablated, and PB MT-ablated) were treated as described above. Laser ablation was performed in CZB medium followed by oocyte incubation in DMEM + HPG for 30 min. Oocytes were fixed and processed for immunostaining with MLKP-1 antibody as described.

### Image acquisition and live-cell imaging

Confocal and super-resolution images were acquired using a Leica SP8 confocal microscope with Lightning module equipped with a 40X, 1.30NA oil-immersion objective. Super-resolution STED images were acquired using a Leica SP8 confocal microscope with τ-STED module equipped with a 93X, 1.30NA glycerol-immersion objective. Image acquisition software was the Leica Application Suite X. For each image, optical z-sections were obtained using 0.5-1 μm step-size with a zoom factor of 2.5-6. Z-series imaging was used to determine the PB/egg sides. Regardless of spindle orientation, the PB is extruded beyond the egg's confines. Therefore, using z-series imaging, the DNA of the PB always appears outward, whereas the DNA of the egg appears inward, regardless of oocyte orientation. Oocytes from experiments involving comparison of intensities or stages were processed on the same day and imaged maintaining laser settings equal across samples.

Live-cell confocal image acquisition was performed using a Leica SP8 confocal microscope system with a 40X, 1.30NA oil-immersion objective, equipped with a heated, humidified stage top incubator with 5% $CO_2$ and 37° C (Tokai Hit, STX stage top incubator). To observe progression through cytokinesis, images of oocytes were acquired every 20 min with 15 optical sections across the entire thickness of each oocyte at 1024 × 1024-pixel image resolution and 600 Hz acquisition speed. For EB3-GFP tracking during cytokinesis, images were taken every 0.5 s at a single plane at 1024 × 512-pixel image resolution and 1000 Hz acquisition speed.

### Cloning and cRNA preparation

Full-length clone of mouse Mklp1 (MGC: 54492) was purchased from Open Biosystems and amplified for insertion into pIVT-eGFP or pIVT-mCherry[59]. Gap43-eGFP was a gift from Dr. Greg FitzHarris (U. of Montreal). To generate cRNA of *Eb3-eGfp*[60], *Mklp1-eGfp*, *Mklp1-mCherry*, and *Gap43-eGfp*, plasmids were linearized and transcribed in vitro using mMessage mMachine T7 or SP6 kit (Ambion, Cat# AM1344) according to manufacturer's protocol.

cRNA was purified using SeraMag Speedbead (Sigma Aldrich, Cat# GE65152105050250) nucleotide purification method previously described[61]. Briefly, in vitro transcription reaction solution was brought up to 150 μl and mixed with 100 μl of magnetic beads and let stand for 5 min. Beads were then pelleted using a magnetic stand and washed with 80% ethanol. cRNA was eluted using 20 μl nuclease-free water and stored at −80° C.

### Image analysis and quantification

All images and videos were analyzed and quantified using Imaris software (9.9.1) (Bitplane, Oxford Instrument Company) and Fiji (2.3.0/1.53f51)[62]. Quantification of volume and intensity were performed by creating a region of interest (ROI) with the surfaces tool in Imaris. To determine ROI, threshold of signal was determined from control groups and applied in treatment groups. For co-localization analyses of MKLP1-CITK and MKLP1-RACGAP1, the co-localization analysis tool in Imaris was used to determine the Manders overlap coefficient, which quantifies the co-incidence of two pixels in different channels within a set threshold[63]. Briefly, each channel was used to determine an ROI, which was used for the co-localization analysis, and the remaining channel's intensity to be measured was set to determine the Manders overlap coefficient.

For EB3-eGFP speed tracking, only cells oriented parallel to the imaging plane were imaged to prevent differences due to angular orientations. The imaging plane was selected based on distinct visualization of the dark zone. Videos were processed by Gaussian filter blend and background subtraction. Individual puncta were then determined using the spots tool and filtering for spots that could be tracked in at least 3 continuous frames. For mapping the directionality of the EB3-eGFP comets, a track path displacement tool was used on Imaris. For EB3-eGFP intensity measurements, the first frame of each video was used to compare the intensity of the egg side to the PB side. The dark zone was used as a reference to distinguish the egg and the PB and mark ROIs.

### Statistical analysis

As indicated in the figure legends, one-way ANOVA and unpaired Student's t-test analyses were performed to examine statistical differences between groups using GraphPad Prism software (9.4.1). $p < 0.05$ was considered statistically significant. All error bars shown reflect standard errors of means.

### Reporting summary

Further information on research design is available in the Nature Portfolio Reporting Summary linked to this article.

## Data availability

No sequence or proteomic data has been generated in this study. All image data supporting the findings of this study are available from the corresponding author upon request. The image analysis source data used for calculations of graphs is available on Figshare [https://doi.org/10.6084/m9.figshare.6025748]. Source data are provided with this paper.

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

## Acknowledgements

The authors thank members of the Schindler lab for helpful discussions, Dr. Jessica Shivas for sharing her expertise and assistance with confocal and super-resolution microscopy, and Dr. Di Wu for guidance with the fluorescence in situ hybridization protocol. The authors also thank Yi Jing Calvin Liu for his help with coding a script for analyzing live-cell imaging data and H. Adam Steinberg, owner of ArtforScience, for creating the schematic models and assisting with figure layout. We acknowledge Dr. Greg FitzHarris (U. Montreal) for sharing the Gap43-eGfp construct. This work was supported by an NIH grant R35 GM136340 to KS. D.L.V. and A.B. were supported by NIH grant R35 GM142537. S.P. and A.R.S. were supported by a grant from the NIH (R35 GM139695-01A1).

## Author contributions

G.J. and K.S., in discussion with ARS, conceived the project. G.J. designed and performed experiments, data analysis, and figure preparation. D.L.V. and A.B. designed and performed laser ablation experiments. G.J. and K.S. wrote and edited the manuscript. S.P., A.R.S., and A.B. provided intellectual feedback and contributed to manuscript editing.

## Competing interests

The authors declare no competing interests.
