## [Peer Review File · Nature Communications]

An oocyte meiotic midbody cap is required for developmental competence in miceEditorial Note: This manuscript has been previously reviewed at another journal that is not operating a transparent peer review scheme. This document only contains reviewer comments and rebuttal letters for versions considered at *Nature Communications*. Mentions of prior referee reports have been redacted.

REVIEWER COMMENTS

Reviewer #1 (Remarks to the Author):

This is a manuscript that describes the cap-like structure made by the central spindle component MKLP1 in the meiotic midbody (mMB) of mouse oocytes. mMB is enriched with ribonucleoproteins including nascent proteins, and its disruption results in the developmental arrest of the parthenogenetically activated egg. New data of laser ablation in the egg and MB side of the spindle is beautiful, and it is convincing that mMB has some functional significance. However, several data interpretations still require more clarifications as listed below.

Major comments:

- Based on the HPG signal and Fig.5f diagram, the nascent protein appears to be in the cap (magenta in 5f) which is located outside of the oocyte to be excised and forms the MB remnant (MBR). The authors describe that the cap ablation leaked the protein content into the PB side, I'm puzzled why that would compromise embryonic development since this structure is already outside of the egg. Ablating the excised material should not alter the oocyte content. In relation to this, the authors propose "inheritance of a translationally active meiotic MB would be critical to produce an egg developmentally competent to support early embryogenesis". However, a new video that tracks MKLP1 and mMB dynamics suggests that mMBR remains outside of the egg after abscission. Therefore, it is not clear if mMBR will be ever inherited in the egg. I find the results are overall interesting but the mMBR inheritance by egg needs to be experimentally shown if they wish to propose this model. Or, they may consider amending their hypothesis since it is also possible that the oocyte may be removing the proteins that interfere with the downstream events, or the abscised mMBR may have another function as an MTOC as it has been shown in mouse embryos. Authors should describe the logic behind their thinking better.

Other comments:

- I am uncertain why authors consider asymmetric cell division should be linked to asymmetric abscission. Do other cells that undergo asymmetric cell division usually undergo asymmetric abscission? If so, how does it actively contribute to asymmetric cell division? Since the word symmetric and asymmetric is not common in the context of abscission, the authors may consider defining the term first. Similarly, the authors used CHMP4B distribution as evidence of symmetric abscission, but is it known that CHMP4B distributes differently in asymmetric abscission? If so, please refer to the appropriate citations.

- Compression of oocytes is known to result in symmetric cytokinesis with developmentally competent daughter cells, while compression also compromised the cap structure of mMB in this study. This part seems contradictory to the author's conclusion that the proper cap structure is critical for embryonic development. Please explain better the logic behind their thinking.

- Nocodazole treatment appears to have a similar effect on compression. Does this treatment induce symmetric cytokinesis? Please describe how oocytes and embryos develop (or do not develop) after washing off the nocodazole.
- Fig. 5a: Please quantify the signal leak level of HPG after nocodazole treatment. Also, the images are small, and difficult to identify the leak. Showing the single channel for HPG might be helpful, while the YZ and XZ images seem less helpful here.
- In both nocodazole treatment and ablation experiments, the leak level of HPG is relatively minor. Authors speculate that the protein leak is the major cause of embryonic failure but does it stop only embryogenesis, or does it actually stop the 2nd meiosis as well? Please explain what will occur to those treated oocytes. Further, the authors should discuss other possible roles of the cap in this process considering the protein leak seems minor.
- Fig. 5f diagram indicates the addition of sperm, which is not what was done in this study, therefore it is misleading. Further, please explain why artificial induction was used here, not the sperm as indicated in the diagram.

Minor comments:

- Depending on the images given in the manuscript, PRC1 (Fig 1c) and MKLP2 (Fig. S2c) also seem to have a cap-like morphology. If this is the angle issue, authors should replace either picture to avoid confusion.
- Line104, “ectopically expressed Mklp1-Gfp in live oocytes (Video S1)”. Is it the typo of “exogenously”?
- Please provide the time stamp and the imaging condition (time interval, z-stack range, and interval) in all videos.
- Please include the PB-MT ablated result in Fig 5e graph.
- It is important to write out the details of chemical treatment experiments in the method section since the treatment length is unusually long. How authors came up with 9-11 hours of treatment for nocodazole and Puro/CHX? Please explain.

Reviewer #2 (Remarks to the Author):

[redacted]

My main concern comes from the statement that abscission is symmetric. I do not think any of the data

proposed here shows this. The live imaging movie suggests that the microtubules are cut on the oocyte side, but it's not obvious what happens on the PB side and the cap remains with the polar body, which could suggest asymmetric abscission. It is also not clear what happens to the membranes. The pictures in figure 2 show actin around the cap, but it is not obvious that it is separated from the midbody- and I also cannot see well the actin cortex of the midbody, so it is also difficult to conclude that the 2 membranes are separated here. The CHMP4B stainings are not incredibly clear, and are also not a proof of abscission, only recruitment. I do not think the symmetric versus asymmetric part of the story is incredibly crucial, so I would suggest to just tone down the claims and say that all this data "suggests that the oocyte undergoes symmetric abscission".

[redacted]. If I understand correctly, the authors suggest that before abscission happens, the cap prevents leakage of translated proteins to the PB. When is translation measured in eg figure 4? If this happens after abscission, what is the role of the cap? I would expect that in that case translated products are just stuck in the midbody remnant. Maybe a cartoon that recapitulates everything would help clarify this contradiction.

Minor concern:

I appreciate that the authors added some live imaging, however it is difficult to see the cap in the figure. Maybe a bigger close-up would help, or a 3D rotation video? I think this is crucial so that the reader is convinced and cannot think the cap is an artefact of fixation.

From Figure 1, it looks like PRC1 is also asymmetric, thus I think the authors should tone down the claim that only Central spindle form the cap. It also makes sense, since the authors suggest that microtubules are responsible for the asymmetry of the cap, and PRC1 is the main microtubule antiparallel bundling protein in the central spindle.

I think the authors should consider the possibility that the deformation of the cap is due to mechanics, since they can revert it by pressing on the embryos. I think it would be interesting to press on the embryo at different time point (not just during cytokinesis) and see whether the shape of the cap changes, although I do realise that without the first author in the lab, that might be complicated. There is a sentence in line 198, but I think this should earlier as the paragraph ending on line 169 feels very conclusive.

Figure 3d should probably show a mean of different examples.

Line 177: I do not think the microtubules are symmetric, especially in the 50 μM example.

Is figure 3a inverted? It looks like the oocyte is on the right.

Figure 3f: the color code is not very much in line with the conclusions. Arrows in white show "motion to the right", whilst the PB is actually on the bottom right. The authors should color code in 4 colours, top left/bottom left/top right/bottom right, which would be a lot more precise (and not a lot more complicated).

Why is the quantification in figure 5c not showing the different controls?

Dear Reviewers,

Enclosed is the revision of our manuscript entitled “**An oocyte meiotic midbody cap is required for developmental competence in mice**” for consideration for publication in *Nature Communications*. We have responded to your concerns and are grateful for your expert insight. The most significant changes you will find are: 1) tempering of abscission conclusions, 2) additional requested analyses, and 3) inclusion of several new schematics and figure labels to highlight conclusions more clearly.

We find that these modifications and adjustments to the text and figure layouts strengthen our manuscript. Below, is a point-by-point response to your queries and suggestions. We have copied and italicized your points, which are followed by our responses below.

Sincerely,

Karen Schindler

Reviewer 1:

1a. Based on the HPG signal and Fig.5f diagram, the nascent protein appears to be in the cap (magenta in 5f) which is located outside of the oocyte to be excised and forms the MB remnant (MBR). The authors describe that the cap ablation leaked the protein content into the PB side, I'm puzzled why that would compromise embryonic development since this structure is already outside of the egg. Ablating the excised material should not alter the oocyte content.

In our ablation experiments, we ablated oocytes in early Telophase I. This was before abscission has occurred and therefore the structure is not outside of the egg. Instead, it is between the egg and the forming or emerging polar body that is not yet a separate cell. We indicate this in the revised text (referring to diagram now 6e):

“In early Telophase I oocytes, a time point before abscission occurs, we employed a multi-photon laser ablation...”

We also moved the Supplemental panel S5a to Figure 5a. This panel shows that the MKLP1 cap is the boundary to HPG labeling of nascent translation in early Telophase. In hindsight, we realized that it strengthens the points in Figures 5 and 6. We have adjusted the text and legends to reflect this change.

1b. In relation to this, the authors propose “inheritance of a translationally active meiotic MB would be critical to produce an egg developmentally competent to support early embryogenesis”. However, a new video that tracks MKLP1 and mMB dynamics suggests that mMBR remains outside of the egg after abscission. Therefore, it is not clear if mMBR will be ever inherited in the egg. I find the results are overall interesting but the mMBR inheritance by egg needs to be experimentally shown if they wish to propose this model. Or, they may consider amending their hypothesis since it is also possible that the oocyte may be removing the proteins that interfere with the downstream events, or the abscised mMBR may have another function as an MTOC as

it has been shown in mouse embryos. Authors should describe the logic behind their thinking better.

We now realize that our logic was not clear in this section. We modify this sentence to read:

“Because oocytes must produce proteins critical for successful meiosis and early embryogenesis, we hypothesized that MBs would locally translate proteins. Furthermore, we hypothesized that the egg would have a mechanism to retain these proteins by preventing their escape into the polar body, a mechanism which could be critical to produce a developmentally competent egg.”

We also mention the alternative model of sequestration in the final discussion:

“We speculate that mMBR release from eggs can later act as signaling organelles during fertilization or pre-implantation embryogenesis if they are phagocytosed by the developing embryo. Alternatively, the mMBR may harbor maternal molecules that are inhibitory to early embryogenesis and therefore need to be sequestered away. Further insight into the identity of the RNA transcripts and proteins in mMBRs is needed to understand their potential roles in embryo development.”

As indicated in response 1a, cell-cycle timing and cap formation timing is key to the experiments and interpretations. The proteins made in the midbody in early Telophase I are retained in the egg by the cap and abscission into a mMBR occurs later (~Met II). Experiments tracking the fate of the mMBR are outside of the scope of this report but are of high priority for our follow up studies. We elected to show the abscission data here because of the gap in knowledge about abscission occurring in oocytes and for a more complete story.

We added a new schematic in Figure 2 (Fig. 2d) that highlights the timing and formation of the structures. We also added new schematic in new Figure 5 (Fig. 5d) that illustrates the timing of translation and the structures. We hope that these new visuals clarify our conclusions for the reviewer and readers.

2a. *I am uncertain why authors consider asymmetric cell division should be linked to asymmetric abscission. Do other cells that undergo asymmetric cell division usually undergo asymmetric abscission? If so, how does it actively contribute to asymmetric cell division?*

We were naïve in our thinking about a correlation of asymmetry of cytokinesis and abscission. Upon closer inspection of the literature, we understand your point, that there is no biological precedent that an asymmetric cell division is associated with abscission on one side of the MB. Therefore, our rationale is confusing. In fact, the important finding that is new to oocyte biology is that abscission occurs. We modified our text to read:

“Meiotic MBR formation and abscission have not yet been evaluated in mouse oocytes. To determine if mMBRs form, we first marked mMBs with anti-MKLP1 in Metaphase II-arrested eggs. The images showed that the mMB left the egg and the resulting mMBR was sandwiched between the egg and the zona pellucida, bound by the egg and PB membranes that were marked by phalloidin-based actin staining (Fig. 2a).”

2b. *Since the word symmetric and asymmetric is not common in the context of abscission, the authors may consider defining the term first.*

We modified our use of symmetric and asymmetric abscission, terminology that we adopted from a review (PMCID: PMC6891101). We now understand that that is not common terminology used in the field and have edited our text throughout.

2c. *Similarly, the authors used CHMP4B distribution as evidence of symmetric abscission, but is it known that CHMP4B distributes differently in asymmetric abscission? If so, please refer to the appropriate citations.*

It is documented that Chmp4b, and other ESCRT III proteins co-localize to sites of microtubule constriction in late cytokinesis where abscission subsequently occurs (for example, PMCID: PMC3064317). In this citation, the authors conclude that there is a “direct cause and effect relationship between Chmp4b recruitment to constriction sites and membrane abscission.” In Guizetti *et al* (PMID: 21310966), the data in Figure 3 beautifully shows two zones of Chmp4b in Hela cells undergoing bilateral abscission. Importantly, in Gromley *et al* (PMID: 16213214), the authors show in Figure 7 that v-SNARE vesicles (which interact with ESCRT III proteins (PMCID: PMC2743992) move to one side of the MB of asymmetrically abscising Hela cells and that unilateral cutting occurs at this site. We now include these citations in the manuscript. However, based on comments from reviewer 2, we now temper the conclusion that there is bilateral abscission and focus on the mMBR formation. We also state:

“Although we did not examine membrane scission and further experimentation is required to confidently establish the number of abscission sites, the data suggest that mMBs are abscised from the egg into mMBRs. Together, the data indicates that a unique cap-containing mMB forms in early Telophase I, the cap resolves in late Telophase I, and a mMBR forms in Metaphase II (Fig. 2d).”

3. *Compression of oocytes is known to result in symmetric cytokinesis with developmentally competent daughter cells, while compression also compromised the cap structure of mMB in this study. This part seems contradictory to the author’s conclusion that the proper cap structure is critical for embryonic development. Please explain better the logic behind their thinking.*

We apologize for the lack of clarity in our thinking. In the manuscript that described the compression method, they evaluated developmental competence of parthenotes from symmetric division. Their results showed (Fig. 3 in Otsuki *et al.* 2012) that there is a significant reduction in parthenote formation, indicating *reduced* developmental

competence. Their finding that there is ~50% reduction of developing to a 4-cell embryo is consistent with our cap ablation consequences.

This is clarified in our revised text which now reads:

To address our hypothesis, we induced symmetric division by gently compressing oocytes at Metaphase I. This method results in two daughter cells of equal size that are viable but have reduced developmental competence²⁹.”

4. Nocodazole treatment appears to have a similar effect on compression. Does this treatment induce symmetric cytokinesis? Please describe how oocytes and embryos develop (or do not develop) after washing off the nocodazole.

The nocodazole and compression experiments were performed at different times in meiotic maturation. The compression occurred at Metaphase I (8h time point). We further clarified this point in the methods section:

“To induce symmetric division of oocytes, cells were compressed at Metaphase I²⁹. Briefly, after culturing for 8 hours (Metaphase I time point),...”

In contrast, nocodazole treatment occurred after Anaphase I onset (11h timepoint; early Telophase I). We did not assess oocytes or embryos after wash out as this is outside of the scope of our experiments. We further clarified the biological timing in the methods section:

“To depolymerize microtubules and actin during mMB formation in early Telophase I, oocytes were cultured in CZB for 11 hours and then transferred to media containing nocodazole...”

For clarity, we also put this information in the results section:

“To test this hypothesis, we perturbed microtubules during mMB formation (early Telophase I) using nocodazole treatment...”

We also put this information as a label in Figure 4b.

5. Fig. 5a: Please quantify the signal leak level of HPG after nocodazole treatment. Also, the images are small, and difficult to identify the leak. Showing the single channel for HPG might be helpful, while the YZ and XZ images seem less helpful here.

This is a great suggestion. We have quantified these images and modified the figure (now Figure 5b-c). The top panels show a single z slice of HPG labeling in gray and zoomed in. The arrows in nocodazole treatment show the streaks of signal coming out of the mMB region. We elected to keep the YZ/XZ in the figure in case other readers may find them useful and for continuity with other figures.

6. *In both nocodazole treatment and ablation experiments, the leak level of HPG is relatively minor. Authors speculate that the protein leak is the major cause of embryonic failure but does it stop only embryogenesis, or does it actually stop the 2nd meiosis as well? Please explain what will occur to those treated oocytes. Further, the authors should discuss other possible roles of the cap in this process considering the protein leak seems minor.*

We do see the beginning of meiosis II completion. This was observed by the bulging of the 2nd polar body. However, to activate eggs, we use Cytochalasin D treatment to internalize the maternal DNA in the PB. Therefore, the 2nd polar body regresses after starting to bulge. We indicate this in the text. Some eggs do become 2C embryos, and we observed a significant deficiency in 2C to 4C cytokinesis. Because some cells became 2C zygotes, these data support that meiosis II was not affected. The text was revised:

“In contrast, ~75% of cap-ablated parthenotes failed to develop past the egg/one-cell stage (Fig. 6d) despite completion of meiosis II as visualized by 2nd polar body bulging.”

7. *Fig. 5f diagram indicates the addition of sperm, which is not what was done in this study, therefore it is misleading. Further, please explain why artificial induction was used here, not the sperm as indicated in the diagram.*

Our intention was to provide a model that summarizes our speculation of the role of the mMB cap *in vivo*, which would require fertilization by sperm. To not be misleading, we have removed the sperm from the image and indicate addition of strontium chloride (SrCl₂) (now Figure 6e). We elected to activate the eggs in the experiment because the timing between ablation and IVF procedures were not compatible. We indicate this in the text:

“We chose to activate eggs as a proxy for fertilization because the timing of ablation procedures and the processing steps required for *in vitro* fertilization were not experimentally compatible.”

8. *Depending on the images given in the manuscript, PRC1 (Fig 1c) and MKLP2 (Fig. S2c) also seem to have a cap-like morphology. If this is the angle issue, authors should replace either picture to avoid confusion.*

We strongly believe that PRC1 and MKLP2 are not part of the cap. The rounded localization reflects the socket shape of the PB side of the mMB, whereas the MKLP1 and RacGap is a more pronounced bulge that go beyond the microtubule socket extending into the PB. In the future, higher resolution microscopy could be used to support this interpretation. We did however replace MKLP2 in figure S2 with an image that is clear that it is not in the cap. We clarify in the text:

“Both proteins sometimes had concave staining patterns on the PB side of the dark zone, tracking with the “socket” shape of the microtubules.”

“... cap-like structure (cap) that surrounded the microtubules on the egg side and always protruded towards the extruding PB, going beyond the socket-shaped microtubules.”

9. *Line 104, “ectopically expressed Mklp1-Gfp in live oocytes (Video S1)”. Is it the typo of “exogenously”?*

Ectopic expression is standard language used in literature involving oocyte injection, but we have edited this term to exogenous to reduce field-specific jargon.

10. *Please provide the time stamp and the imaging condition (time interval, z-stack range, and interval) in all videos.*

We have provided this information in all the videos.

11. *Please include the PB-MT ablated result in Fig 5e graph.*

We apologize for the lack of clarity. When we did the PB-MT ablation, this was a control to show that ablation specifically has to occur at the cap to disrupt MKLP1 and that the ablation procedure itself doesn't disrupt it. In doing the HPG experiment, we simplified our controls to only include egg-side MT ablation since they appeared the same and we decided that the egg side MT was the most important of the 2 controls. This simplification allowed us to increase the number of oocytes we could examine in a single experiment. We therefore do not have the data to include in a revised graph. Because the cap is intact in the PB-side ablation, it is highly unlikely there would be any HPG leakage. For clarity, we edited the text to indicate this:

“Early Telophase I oocytes were either not exposed to laser ablation (non-ablated controls) or exposed to a multi-photon laser ablation in the cytoplasm (cytoplasmic ablation), at the egg side of the spindle (egg MT-ablated) or at the mMB cap (to disrupt its integrity); we excluded the PB-side MT ablation in this experiment.”

12. *It is important to write out the details of chemical treatment experiments in the method section since the treatment length is unusually long. How authors came up with 9-11 hours of treatment for nocodazole and Puro/CHX? Please explain.*

The nocodazole treatment was performed for 30 min and is indicated in the methods. The 9h culture occurred without inhibitors to get oocytes to anaphase onset. Drugs were then added. When we tried short exposures such as 30 min, 1h, 1.5h, 2, 2.5h, 3h, 3.5h, and 4h treatments, we did not see effects. At this point the time course gets tricky because the timing is in the middle of the night. We then found that Puro/CHX treatments arrested oocytes in Telophase I, essentially synchronizing them. Because they all arrested at the same stage, oocytes could remain in the drug and then fixed the next morning. Importantly, this workflow also allowed for the experimentalist to maintain essential, healthy sleep habits without compromising the experiment.

Reviewer 2

1. *My main concern comes from the statement that abscission is symmetric. I do not think any of the data proposed here shows this. The live imaging movie suggests that the microtubules are cut on the oocyte side, but it's not obvious what happens on the PB side and the cap remains with the polar body, which could suggest asymmetric abscission. It is also not clear what happens to the membranes. The pictures in figure 2 show actin around the cap, but it is not obvious that it is separated from the midbody- and I also cannot see well the actin cortex of the midbody, so it is also difficult to conclude that the 2 membranes are separated here. The CHMP4B stainings are not incredibly clear, and are also not a proof of abscission, only recruitment. I do not think the symmetric versus asymmetric part of the story is incredibly crucial, so I would suggest to just tone down the claims and say that all this data "suggests that the oocyte undergoes symmetric abscission".*

We thank the reviewer for their guidance and explanation. We have extensively modified this section that is retitled "Meiotic midbody remnant formation" to temper the conclusions about bilateral abscission and highlight limitations of data interpretation.

2. [redacted]

Yes, this is our working model.

When is translation measured in eg figure 4? If this happens after abscission, what is the role of the cap? I would expect that in that case translated products are just stuck in the midbody remnant. Maybe a cartoon that recapitulates everything would help clarify this contradiction.

Translation was measured in early Telophase I, when the cap is present and ~1.5-2h before abscission. The cap regresses in late Telophase I. We agree, and expect that any proteins made in later Telophase may end up in the mMBR. We clarify this timing in the text:

"Therefore, we investigated whether the mMB of early Telophase I oocytes (pre-abscission) have RNP characteristics.."

We also have created a new figure 2d that summarizes figures 1 and 2 in a temporal manner and new figure 5d that summarizes the cap and translation. We also have carefully labeled all figures and increased some zoom insets to highlight nuances in the timing and findings that might not be clear to all readers.

3. *I appreciate that the authors added some live imaging, however it is difficult to see the cap in the figure. Maybe a bigger close-up would help, or a 3D rotation video? I think this is crucial so that the reader is convinced and cannot think the cap is an artefact of fixation.*

To keep oocytes alive, we have to image with lower resolution and larger z steps. Therefore, the cap was only viewed in 1 z-slice of the stack and we cannot make a 3D rotation video. Instead, we now provide larger zooms of cap portion in Figure 1j. We focused on time points in early Telophase I where the bulge of the cap is obvious in the video and one in late Telophase where cap regresses .

4. *From Figure 1, it looks like PRC1 is also asymmetric, thus I think the authors should tone down the claim that only Centralspindle form the cap. It also makes sense, since the authors suggest that microtubules are responsible for the asymmetry of the cap, and PRC1 is the main microtubule antiparallel bundling protein in the central spindle.*

Reviewer 1 had this same concern. However, we strongly believe that PRC1 and MKLP2 are not part of the cap. The rounded localization reflects the socket shape of the PB side of the mMB, whereas the MKLP1 and RacGap is a more pronounced bulge that goes beyond the microtubule socket extending into the PB. In the future, higher resolution microscopy could be used to support this interpretation. We did however replace MKLP2 in figure S2 with an image that is clear that it is not in the cap. We clarify in the text:

“Both proteins sometimes had concave staining patterns on the PB side of the dark zone, tracking with the “socket” shape of the microtubules.”

“ ... cap-like structure (cap) that surrounded the microtubules on the egg side and always protruded towards the extruding PB, going beyond the socket-shaped microtubules.”

We also tone down that just Centralspindlin is in the cap, as suggested.

5. *I think the authors should consider the possibility that the deformation of the cap is due to mechanics, since they can revert it by pressing on the embryos. I think it would be interesting to press on the embryo at different time point (not just during cytokinesis) and see whether the shape of the cap changes, although I do realise that without the first author in the lab, that might be complicated.*

We include a new line in the text:

“Alternatively, the cap itself may be sensitive to changes in pressure and other mechanical perturbations, possibilities that could be further evaluated.”

6. *There is a sentence in line 198, but I think this should earlier as the paragraphe ending on line 169 feels very conclusive.*

As suggested, we moved this line up to the end of the preceding paragraph.

7. *Figure 3d should probably show a mean of different examples.*

We now provide a revised figure 3d showing the mean of the examples using a semi-transparent fill style. The data shows that in control, DMSO treated oocytes have 3 peaks of MKLP1 which are the 2 sides of the ring and the cap. In nocodazole treated oocytes, there are 2 peaks of MKLP1 which are the 2 sides of the cap. We edited the text to highlight this finding better:

“Analysis of MKLP1 pixel intensity across the midbody bridge in DMSO controls, showed that there were three peaks of MKLP1 staining. These peaks corresponded to the two sides of the ring and the cap. Upon 50 μ M nocodazole treatment, only two peaks of MKLP1 staining were apparent, corresponding to the sides of the ring (Fig. 3d) and a cap peak was not detected.”

8. Line 177: *I do not think the microtubules are symmetric, especially in the 50 microM example.*

We apologize for the lack of clarity and use of the word symmetric. What we mean here is that the ball and socket shapes of the MTs are gone. We modified the text to read:

“Notably, at these doses, microtubules were still present, but the ball-and-socket morphology of the microtubules disappeared.”

9. *Is figure 3a inverted? It looks like the oocyte is on the right.*

We reviewed the files from this image and confirm that it is not inverted. The PB looks larger in this image because of the z-planes chosen to best highlight the cap and because more of the PB is showing than the egg.

10. *Figure 3f: the color code is not very much in line with the conclusions. Arrows in white show “motion to the right”, whilst the PB is actually on the bottom right. The authors should color code in 4 colours, top left/bottom left/top right/bottom right, which would be a lot more precise (and not a lot more complicated).*

We realize that our text was not clear because we wrote that EB3 was directed toward the PB, when we meant mMB. We have edited the text and we elect to keep the figure with 2 colors, and hope that the reviewer agrees with our explanation.

11. *Why is the quantification in figure 5c not showing the different controls?*

Figure 5c was a control experiment conducted to show precision of ablation. The goal was to show that the cap was only perturbed when the cap was targeted and not when MT on either side of the cap were ablated. Experiments activating ablated eggs and labeling HPG were done after this control experiment. To ensure that we had sufficient numbers of oocytes to power these experiments, we chose the best control to include—this being the egg side MT ablation control. We therefore cannot include PB side MT analysis for these experiments. We clarify this in the text:

“Early Telophase I oocytes were either not exposed to laser ablation (non-ablated controls) or exposed to a multi-photon laser ablation in the cytoplasm (cytoplasmic ablation), at the egg side

of the spindle (egg MT-ablated) or at the mMB cap (to disrupt its integrity); we excluded the PB-side MT ablation in this experiment.”

REVIEWERS' COMMENTS

Reviewer #1 (Remarks to the Author):

The authors addressed all questions raised by this reviewer.

This is just for advice; For Fig 2c images, adding a blow-up view with a bright field or co-staining images with CHMP4B might clarify better the exact location of mMBR. It is a bit hard to locate where they are with the current resolution.

Reviewer #2 (Remarks to the Author):

I am happy with the answers to my comments. It is a very interesting study and I am glad to recommend acceptance.

Response to reviewers

We thank the reviewers for taking the time to consider our revised manuscript entitled “An oocyte meiotic midbody cap is required for developmental competence in mice.” We were pleased that the reviewers were enthusiastic about our work. We will take Reviewer 1’s advice about how to improve Chmp4b studies for the future.